# Recognition of Japanese university students one year after the discharge of treated water from the Fukushima Daiichi Nuclear Power Station

Hitomi Matsunaga[1]*, Isamu Amir[2], Aizhan Zabirova[1], Yuya Kashiwazaki[1], Makiko Orita[1], Thierry Schneider[3], Masaharu Tsubokura[2], Noboru Takamura[1]

1 Department of Disaster Resilience and Science, Atomic Bomb Disease Institute, Nagasaki University, 1-12-4 Sakamoto, Nagasaki, Japan, 2 Department of Radiation Health Management, Fukushima Medical University School of Medicine, Hikarigaoka-1, Fukushima City, Fukushima, Japan, 3 French Authority for Nuclear Safety and Radiation Protection (ASNR), BP 17, Fontenay-aux-Roses Cedex, France

* hmatsu@nagasaki-u.ac.jp

## Abstract

The study clarified the perceptions of the Japanese university students regarding their acceptance of the discharge of treated water (DTW) into the Pacific Ocean approximately a year after the process began. Among the 1453 study participants, 1246 (85.7%) showed DTW acceptability, and 207 (14.3%) were unacceptable. Compared to non-acceptance group, majoring in science, experience collecting information or knowledge about DTW, the ability to explain the difference between contaminated water and treated water, the belief that the Japanese government provides accurate information about DTW, the belief that decision-making of the Japanese public is calm and rational about the DTW, and having no feel reluctant to consume kinds of seafood in Fukushima were independently associated with the acceptance group. Furthermore, the most common way to collect information about DTW was via television or newspapers. The paper suggested that effectively conveying information about DTW to the younger generation is best done passively, such as through TV, street flyers, or Internet advertisements. Participants who had received some kind of information about DTW were more likely to accept DTW than those who had not. The health and environmental effects of DTW from FDNPS are limited; therefore, this complex issue must be dealt with calmly.

## Introduction

On March 11, 2011, the east coast of Japan's Tohoku region experienced an earthquake of magnitude 9.0 earthquake that triggered a destructive tsunami. Consequently, the cooling system of the Tokyo Electric Power Company Holdings (TEPCO), Fukushima Daiichi Nuclear Power Station (FDNPS) was destroyed, causing the reactor cores to overheat and nuclear meltdown [1]. Since the disaster, the FDNPS has been pumping water to cool the fuel debris of the reactors,

**Data availability statement:** All relevant data are within the paper and its Supporting Information files.

**Funding:** This research was performed by the commissioned research fund provided by F-REI (JPFR25050501). The funders had no role in study design, data collection and analysis, decision to publish, or preparation of the manuscript.

**Competing interests:** No.

generating contaminated water. Groundwater and rainwater that seep into the reactor and turbine buildings also mix with this contaminated water, thereby generating new contaminated water every day [2]. The amount of contaminated water generated was approximately 140 $m^3$/day (the average in 2020) due to multilayered contamination control measures, whereas it was approximately 540 $m^3$/day (May 2014) before the countermeasures were implemented. Contaminated water was purified using multi-nuclide removal equipment (ALPS), and its storage was continued in a tank within the FDNPS facility. By October 31, 2024, the total amount of ALPS-treated water was 1,293,846 $m^3$, which was 94% of the total storage tank capacity [3]. This paper clearly distinguishes between the terms contaminated water and treated water. Contaminated water is not released into the ocean. Treated water is released into the ocean, which is purified by ALPS and then further diluted with seawater to a level that does not exceed the standard level of radioactivity.

To continue the decommissioning process, the Japanese government and TEPCO have started regular discharge of treated water (DTW) since August 2023. However, the DTW from FDNPS has not been readily accepted either domestically or globally, even after ALPS purification, and has cleared strict safety standards [4, 5]. A particular concern has been raised by neighboring Asian countries, such as China, Korea, and Taiwan. China banned the import of all Japanese seafood and seafood products when DTW began, leading to annual economic losses of 87.1 billion yen (46.7 billion yen from scallops, 7.9 billion yen from sea cucumbers, and 4 billion yen from bonito and tuna) [6]. Issues with DTW are not only people's concerns but have also led to harmful damage and serious problems regarding reputation. Since the start of the DTW process, the Japanese government has cooperated with international organizations and declared its safety. Cooperation with the International Atomic Energy Agency has been strong, and the agency has released comprehensive information on radiation safety [7].

However, the public perceives that social issues associated with nuclear disasters remain a rigorous stance. In a public opinion poll conducted before the DTW, 24% residents of Fukushima accepted it, while the public of Japan was only 18%. [8]. On the other hand, it has been clarified that concerns about DTW were decreasing after it had begun [9,10]. The younger generation of Japanese emphasized sustainability, safety, and international responsibility of DTW, and it was reported that they were prone to be skeptical of nuclear energy policy [11]. In contrast to older generations, who are primarily concerned with economic impacts and damage to fisheries, younger generations tend to focus on long-term environmental sustainability and the effects on marine ecosystems by DTW [12]. In this study, we confirmed the perceptions of the Japanese university students regarding the acceptance of DTW a year after its initiation. DTW into the ocean is a long-term issue that lasts approximately 30 years or more. Therefore, promoting an understanding of DTW among the younger generation is important, as they will be responsible for the future.

## Methods

### Study participants

The survey was conducted from June to July 2024, about a year after the initiation of DTW. The survey was conducted among students of all years at four universities: two in Fukushima Prefecture and two outside prefectures. Two universities outside Fukushima Prefecture were selected, each located in a prefecture facing the sea on an island different from Honshu, where Fukushima is situated. Students from all faculties were surveyed, including those in the Medicine (sixth-year course), Pharmacy, Engineering, Information and Data Science, etc. (S1 Table). The questionnaire was created in both paper and digital formats, with a QR code included, allowing respondents to choose their preferred method: paper or scanning the QR code. Data were collected after school or before the lecture (not about treated water). The purpose, methods, and ethical considerations of the study were explained using a leaflet, and responses to the questionnaire provided informed consent. Each university in Fukushima received 174 (out of approximately 300 distributed) and 387 (out of approximately 600 distributed) responses, with an additional 772 (out of approximately 1000 distributed) and 453 (out of approximately 600 distributed) received from outside Fukushima. The total number of responses was 1786 (including 80 missing values). This study aimed to investigate the risk perception of DTW. Therefore, students who answered that they were unaware of DTW in the FDNPS were excluded from the analysis. Among the 1706 valid responses, 253 (14.8%) were unaware of DTW. A total of 1453; 117 (valid response rate: 67.2%) and 368 (valid response rate: 95.1%) were in Fukushima, and 631 (valid response rate: 81.7%) and 337 (valid response rate: 74.4%) were outside Fukushima were regarded as valid after excluding incomplete responses. All processes in this study were reviewed and approved by the Ethics Committee of the Nagasaki University Graduate School of Biomedical Sciences (No.23081803).

### Questionnaire

The questionnaire was formulated based on previous research conducted in an area affected by the FDNPS accident and the recognition of DTW [13] (S2 Table). The questionnaire encompassed inquiries on demographic characteristics such as sex, age, major of university, grade, and living area. Regarding DTW, respondents were also asked to answer Yes or No regarding whether they had experience collecting information or knowledge about DTW and whether they were able to explain the difference between contaminated water and treated water. To answer how they obtained information or knowledge about DTW, they answered a multiple-choice question and selected Lecture at a University or Workshop, Internet, Television (TV) or Newspaper, and Others. We categorized eight methods of information collection based on the results as Lecture only, Internet only, TV/Newspaper only, Lecture and Internet, Lecture and TV/Newspaper, Internet and TV/Newspaper, All Sources (Lecture, Internet, and TV/Newspaper), and No experience. Regarding the acceptance of DTW, we asked, "Do you accept the ocean discharge of treated water from FDNPS?" Further, we asked whether the decision-making of the Japanese public is calm and rational about the DTW, and whether they feel reluctant to consume kinds of seafood in Fukushima, responded "yes," "probably yes," "probably no," or "no". Regarding the Japanese government providing accurate information about DTW, they were asked to choose between three options: yes, no, or unjudged. Participants were also asked to choose the most concerning impact of DTW on human health, ocean environment, marine creatures, societal issues such as reputational damage, and others.

### Statistical analysis

Age was categorized based on the distribution into two groups: <21 years and ≥ 21 years (S3 Table). The majors were categorized into science and humanities for analysis (S1 Table). The chi-squared test was used to clarify the differences between the acceptance of DTW and each variable. The question item regarding the Japanese government providing accurate information about DTW was analyzed only using yes or no, excluding "unjudged." Furthermore, the chi-square test was performed on the differences between each method of collecting information and the acceptance of DTW. A

logistic regression analysis was applied to the "Acceptance" reference group to clarify the independent association with various factors. The items in the logistic regression analysis were selected from those that resulted in statistically significant differences in the chi-squared test for acceptance. Data analyses were performed using IBM SPSS Statistics version 29, and statistical significance was set at $p < 0.05$.

## Results

Table 1 shows the chi-square test results for sociodemographic factors, recognition of DTW, and concerns about seafood versus acceptance of DTW. Of the 1,453 responding students, when asked about acceptance of DTW from the FDNPS, 719 (49.5) answered Yes, 527 (36.2) answered probably yes, 158 (10.9) answered probably no, and 49 (3.4) answered

**Table 1. Comparisons of each factor accepting DTW from the FDNPS.**

| | Response | Overall (N = 1453) n (%) | Accept 1246 (85.7) n (%) | Do not accept 207 (14.3) n (%) | *p*-value |
|---|---|---|---|---|---|
| Living areas | Inside Fukushima<br>Outside Fukushima | 485 (33.4)<br>968 (66.6) | 427 (34.3)<br>819 (65.7) | 58 (28.0)<br>149 (72.0) | 0.077 |
| Sex | Female<br>Male | 680 (46.8)<br>773 (53.2) | 564 (45.3)<br>682 (54.7) | 116 (56.0)<br>91 (44.0) | 0.004 |
| Age (years) | < 21<br>≥ 21 | 821 (56.5)<br>632 (43.5) | 692 (55.5)<br>554 (44.5) | 129 (62.3)<br>78 (37.7) | 0.068 |
| Major | Science<br>Humanities | 1211 (83.3)<br>242 (16.7) | 1050 (84.3)<br>196 (15.7) | 161 (77.8)<br>46 (22.2) | 0.020 |
| Grade | 1st<br>2nd<br>3rd<br>4th<br>5th<br>6th | 424 (29.2)<br>316(21.7)<br>328 (22.6)<br>247 (17.0)<br>67(4.6)<br>71(4.9) | 353 (28.3)<br>265 (21.3)<br>289 (23.2)<br>211 (16.9)<br>60(4.8)<br>68(5.5) | 71 (34.3)<br>51 (24.6)<br>39 (18.8)<br>36 (17.4)<br>7 (3.4)<br>3 (1.4) | 0.470 |
| The experience of collecting information or knowledge about the DTW | Yes<br>No | 959 (66.0)<br>494 (34.0) | 862 (69.2)<br>384 (30.8) | 97 (46.9)<br>110 (53.1) | <0.001 |
| Able to explain the difference between contaminated water and treated water | Yes<br>No | 603 (41.5)<br>850 (58.5) | 556 (44.6)<br>690 (55.4) | 47 (22.7)<br>160 (77.3) | <0.001 |
| The accuracy of information about DTW provided by the government of Japan | Yes<br>No | 553 (38.1)<br>331(22.8) | 532 (42.7)<br>263 (21.1) | 21 (10.1)<br>68 (32.9) | <0.001 |
| | Unjudged | 569 (39.2) | 451 (36.2) | 118 (57.0) | - |
| The calm decision about DTW in public opinion | Yes<br>Probably Yes<br>Probably No<br>No | 110 (7.6)<br>389 (26.8)<br>697 (48.0)<br>257 (17.7) | 104 (8.3)<br>351 (28.2)<br>574 (46.1)<br>217 (17.4) | 6 (2.9)<br>38(18.4)<br>123 (59.4)<br>40 (19.3) | <0.001 |
| The most concerning impact of the DTW | human health<br>ocean environment<br>marine creatures<br>social issues<br>others | 487 (33.5)<br>212 (14.6)<br>266 (18.3)<br>472 (32.5)<br>16 (1.1) | 388 (31.1)<br>171(13.7)<br>221 (17.7)<br>450 (36.1)<br>16 (1.3) | 99 (47.8)<br>41 (19.8)<br>45 (21.7)<br>22 (10.6)<br>0 (0) | <0.001 |
| Feel reluctant to consume seafood from Fukushima | Yes<br>Probably Yes<br>Probably No<br>No | 70 (4.8)<br>237(16.3)<br>416 (28.6)<br>730 (50.2) | 33 (2.6)<br>175 (14.0)<br>349 (28.0)<br>689 (55.3) | 37 (17.9)<br>62 (30.0)<br>67 (32.4)<br>41 (19.8) | <0.001 |

Chi-square test; FDNPS, Fukushima Daiichi Nuclear Power Station; DTW, discharge of treated water.

no. Hence, we defined 1246 (85.7%) as accepting and 207 (14.3%) as not accepting DTW a year after the initiation of DTW from the FDNPS. The proportion of males (682, 54.7%) who accepted DTW was higher than that of females (564, 45.3%) (p = 0.004). Further, the major of science was significantly different between accepted DTW (1050, 84.3%) and those that did not (161, 77.8%) (p = 0.020). The living area, age, and grade were not significantly different between the groups that accepted DTW and those that did not. The proportion of students who accepted DTW was higher among those who had experience in collecting information or knowledge about DTW (69.2% vs. 46.9%; p < 0.001). Overall, 603 (41.5%) respondents answered that they were able to explain the differences between contaminated and treated water. The group that could explain the difference also had a higher percentage of acceptance (44.6% vs. 22.7%; p < 0.001). 553 (38.1%) respondents believed that the government provides accurate information about DTW, 331 (22.8%) respondents answered no, and 569 (39.2%) were unjudged. Compared with Yes and No answers, those who believed the Japanese government provides accurate information were more likely to accept DTW than those who did not (42.7% vs. 21.1%; p < 0.001). In total, 34.4% (499) of the students thought that the public made a calm and rational decision about DTW from the FDNPS. The students who accepted DTW had a significantly higher percentage of respondents who considered the public reaction to DTW calm (36.5% vs. 21.3%; p < 0.001). The respondents thought that the most concerning impact of DTW was human health (487; 33.5%), followed by societal issues (472; 32.5%), marine creatures (266; 18.3%), and ocean environment (212; 14.6%). Overall, 21.1% (307) had concerns about consuming seafood from Fukushima, and greater numbers in the non-acceptance group than in the acceptance group had significantly greater concerns about consuming seafood from Fukushima (16.6% vs. 47.9%; p < 0.001).

Table 2 summarizes the sources of information and knowledge among students who had experienced collecting information about DTW. The best common way to obtain information about DTW was through TV or newspaper (243;16.7%), followed by lecture only (206;14.2%), Internet and TV or newspaper (198;13.4%), and Internet only (143;9.8%). The group with experience collecting information from the three sources of Internet, TV, and Newspaper, and the "no experience" group showed a significant difference between the acceptance and non-acceptance groups.

Table 3 showed logistic regression analysis confirmed that those who had not felt reluctant to consume seafood from Fukushima (odds ratio [OR]=3.742, 95% confidence interval [CI]: 2.699–5.188, p < 0.001), those who believed the public had made a calm decision about DTW (OR=1.835, 95% CI: 1.259–2.675, p = 0.002), those who were able to explain the difference between contaminated water and treated water (OR=1.832, 95% CI: 1.331–2.806, p < 0.001), those who believed the Japanese government provides accurate information about DTW (OR=1.684, 95% CI: 1.366–2.076, p < 0.001), those who had experience of collecting information or knowledge about DTW (OR= 1.554, 95%CI: 1.118–2.159, p = 0.001), and those who major of science (OR= 1.505, 95% CI: 1.016–2.227, p = 0.041) were independently

**Table 2. Acceptance of DTW regarding the differences in collecting information.**

| | | Overall (N = 1453) % (n) | Accept 85.7(1246) % (n) | Do not Accept 14.3 (207) % (n) | *p*-value |
|---|---|---|---|---|---|
| Experienced methods for collecting DTW information | Only Lecture | 206 (14.2) | 186 (14.8) | 22 (10.6) | 0.114 |
| | Only Internet | 143 (9.8) | 127 (10.2) | 16 (7.7) | 0.271 |
| | Only TV/Newspaper | 243 (16.7) | 218 (17.5) | 25 (12.1) | 0.053 |
| | Lecture and Internet | 56 (3.9) | 53 (4.3) | 22 (1.4) | 0.052 |
| | Lecture and TV/Newspaper | 34 (2.3) | 30 (2.4) | 4 (1.9) | 0.675 |
| | Internet and TV/Newspaper | 198 (13.4) | 176 (14.1) | 18 (8.7) | 0.033 |
| | All (Lecture, Internet and TV/Newspaper) | 82 (5.6) | 73 (5.9) | 9 (4.3) | 0.383 |
| | No experience | 495 (34.1) | 385 (30.9) | 110 (53.1) | <.001 |

Response Yes, TV; television.

**Table 3. Independently factors associated with acceptance of DTW from FDNPS.**

| | | OR (95% CI) | p-value |
|---|---|---|---|
| Sex | Female/Male | 0.892 (0.646–1.232) | 0.488 |
| Feel reluctant to consume seafood from Fukushima | No/Yes | 3.742 (2.699–5.188) | <0.001 |
| The calm decision about DTW in public | Yes/No | 1.835 (1.259–2.675) | 0.002 |
| Able to explain the difference between contaminated water and treated water | Yes/No | 1.832 (1.331–2.806) | <0.001 |
| The accuracy of information about DTW provided by the government of Japan | Yes/No | 1.684 (1.366–2.076) | <0.001 |
| The experience of collecting information or knowledge about DTW | Yes/No | 1.554 (1.118–2.159) | 0.001 |
| Major | Science/Humanities | 1.505 (1.016–2.227) | 0.041 |

Logistic regression analysis, reference; acceptance of DTW from FDNPS, OR; odds ratio, CI; confidence interval, FDNPS; Fukushima Daiichi Nuclear Power Station, DTW; discharge of treated water.

associated with the acceptance of DTW. Sex had a weaker effect than other factors and showed no independent association with this model.

## Discussion

The study confirmed the recognition of Japanese university students one year after the launch of DTW from FDNPS. Furthermore, we evaluated whether their information sources of DTW and its impact on DTW acceptance. Most of the students we surveyed (85.2%;1453/1706) knew about the DTW process, which could be regarded as a high awareness of the various social issues. The research was conducted approximately one year after the DTW process began. Following Japan's announcement of a basic policy of discharging ALPS-treated water into the Pacific Ocean in April 2021 [14], information about treated water had spread to the public through various sources, including those lacking scientific evidence [15]. In particular, just before and after the DTW process began, there was a great deal of media information in Japan's domestic press, which might have driven the public's interest. This result could be regarded as the baseline of recognition and interest levels among younger people after the process and could be used to connect future longitudinal studies. Interestingly, 85.7% of the study participants indicated acceptance of DTW. This distribution has the highest acceptance yet reported. Acceptance has been increasing over the years, as 6.7% agreed with DTW in 2017 (before DTW was initiated) and 32.2% in 2023 (after DTW was initiated), according to a survey of domestic fishermen. The opinions of laypeople regarding radiation have changed depending on their public interest and concerns [16]. The results of the present study suggest that younger generation tends to be more accepting of DTW than older generations. We also found confirmed sex differences regarding DTW, as females tended to be less accepting of DTW than males. The tendency of a correlation between sex and anxiety has already been clarified in previous studies regarding radiation issues, as females tend to have more anxiety and greater refusal of radiation dangers than males [8,17]. The findings of the study also indicate a similar trend. Regarding sex, a significant difference was observed only in the chi-square test. Sex was considered to be a small independent effect on DTW acceptance compared to other factors in the logistic regression model of the study, although sex is one of the important factors in the radiation risk estimation.

Previous studies showed that the most common way that the younger generation gathered information about the disaster was through the Internet [8,18]. However, in the present study, the most prominent way to gather information about DTW among the participants was through TV or newspapers, even though the survey target was younger generations. The Internet depends on the willingness to search; therefore, only a few students may have consciously tried to research DTW. Furthermore, only 14.2% of the students attended a lecture on DTW, and no significant differences were noted between those who attended the lecture and those who did not. Nevertheless, an important finding of this study was that those who had received information about DTW were more likely to accept it than were those with no experience. This result suggests that effectively conveying information about DTW is best done passively, such as through TV, street flyers,

or the Internet through advertising. Providing information through lectures is important; however, it is also necessary to develop a system that can stimulate interest. The findings imply that mass communication is more effective than formal educational interventions, providing critical evidence for leveraging passive media in outreach strategies aimed at younger generations.

Approximately 40% of the participants could explain the difference between contaminated and treated water, and this is important to accept DTW. The water discharged from the FDNPS removed almost all radionuclides on specific filters (via the advanced liquid processing system; ALPS) to levels below the standard value based on the International Commission on Radiological Protection (ICRP) recommendations. Therefore, the treated water discharged from FDNPS does not include nonstandard radionuclides and is quite different from contaminated water that exceeds safety limits. People, groups, and organizations who criticize DTW tend to use the term contaminated water rather than treated water [19,20]. Expanding the information about DTW from FDNPS, which does not exceed the safe levels of standard radionuclides might be effective and could decrease concerns about the radiation effects of DTW. The greatest anxiety regarding DTW is related to its potential health effects, although the risk is scientifically negligible [21]. However, some concerns have been raised about the consumption of seafood from Fukushima, which has led to international issues [22,23]. Notably, the results found that those with low acceptance were most concerned about the human health impact of DTW, whereas those who accepted DTW were most concerned about societal impacts, such as reputational damage caused by DTW. A better understanding of DTW might be fostered by providing a clear explanation of its effects on health.

Overall, 65.7% of the students thought that public opinion regarding DTW was not calm. Nevertheless, a high percentage of students who accepted DTW believed that the public had a calm decision about DTW. Furthermore, logistic regression analysis revealed that trust in the information provided by the government had a positive association with acceptance of DTW. These results suggest that trust in the Japanese government and Tokyo Electric Power Company Holdings is related to the acceptance of DTW. This relationship between trust and information on DTW suggests the influence of political factors. The analysis results of social network information, such as X (formerly Twitter), showed that those in support of DTW were also those who supported the government, whereas those who opposed DTW consisted of groups that criticized the government. The same situation has arisen in Korea, which supports DTW as a country, and most opposition to DTW was to the government [24]. The support of DTW and its risk perceptions suggests a connection with political factors. Regardless, having the trust of laypeople is essential for an organization or nation.

This study had some limitations. First, sampling bias was present, as the data were collected from only four universities inside and outside Fukushima. The study analysis used a combination of data according to the study purpose, but each university student might have had a different recognition of the DTW. Furthermore, because the survey was conducted within a single country, the sampling frame may restrict the international generalizability of the results. Expanding the sampling strategy to include multiple regions and implementing longitudinal follow-ups would help capture temporal changes in public attitudes and enhance the findings' broader applicability. These methodological advancements should be considered in future research to build on the foundations established by the present study. Second, this study also had a sampling bias regarding academic majors. The proportion of respondents from the natural sciences and humanities in our sample does not reflect the national distribution of graduate students in Japan. The results indicate that students majoring in science-related fields were more likely to accept DTW than those majoring in the humanities. However, we were unable to determine whether this was due to their educational background or their interest stemming from a specialized background. Furthermore, the lecture on DTW that the students attended did not evaluate the content, timing, or level of mastery of that lecture. The difference observed between natural science and humanities majors should be interpreted cautiously, given the uneven sample distribution. The acceptance of DTW cannot be understood solely through a natural-science lens, and future studies should intentionally design sampling strategies to include a more balanced representation of humanities and social-science students or other populations with diverse epistemological backgrounds. Third, the research timing was a limitation. In this study, the initial number of valid participants was 1,706; however, in the final

analysis, 235 students (approximately 14%) who were unaware of DTW were excluded from the study. It should be noted that these findings were based on data collected exactly one year after the DTW into the ocean, during a period when people in various positions, both domestically and internationally, including students, were paying attention to DTW from FDNPS. The perceptions of university students immediately after the FDNPS accident or ten years later could be considerably different. Fourth, the questionnaire items were limited. We did not include items such as economic status (e.g., whether students had part-time jobs), residential status (e.g., whether they lived in a dormitory), or the frequency of interactions with friends and family. Regarding information sharing, factors such as the social environment and interpersonal relationships may have influenced the acceptance of DTW. Furthermore, the survey did not assess procedural fairness, stakeholder involvement, or perceived legitimacy of the decision-making process, and the findings reflect only one dimension of acceptance of scientific safety–related perceptions. Therefore, future studies should incorporate questions on social fairness, international justice, and stakeholder engagement to more comprehensively evaluate acceptance of DTW. Moreover, Scientific literacy was not directly assessed in this study. Differences in participants' scientific literacy may have influenced their understanding of technical content related to radiation, environmental monitoring, and safety regulations. Incorporating a validated measure of scientific literacy would strengthen future investigations. Finally, there is the issue of research design. This survey employed a conventional cross-sectional design and relied primarily on descriptive statistics and chi-square tests to compare the acceptance and non-acceptance of DTW by FDNPS. While this approach was appropriate for obtaining baseline data in the local context, the methodological framework may limit the findings' generalizability. Future studies could benefit from incorporating more advanced analytical approaches, such as conceptual modeling or structural equation modeling, to clarify the causal pathways and improve theoretical robustness.

## Conclusion

Overall, 85.2% of the study cohort knew about DTW, and among them, 85.7% expressed acceptance. Students who had received some kind of information about DTW were more likely to accept DTW than those with no experience who got the information. Acceptance was significantly related to trust in the information provided by the government of Japan, although they thought that public opinion regarding DTW was not calm. The health and environmental effects of DTW from FDNPS are limited; therefore, this complex issue must be dealt with calmly

## Supporting information

**S1 Table. This is Major of the study participants.**
(DOCX)

**S2 Table. Questionnaire English-DTW.**
(DOCX)

**S3 Table. Age of the study participants.**
(DOCX)

## Acknowledgments

We would like to thank all study participants, Professor Shoichi Arai, Professor Shin Kawai, and Professor Akiko Fukui for their contributions to the survey.

## Author contributions

**Conceptualization:** Isamu Amir, Aizhan Zabirova.

**Investigation:** Yuya Kashiwazaki.

**Methodology:** Isamu Amir, Aizhan Zabirova.

**Supervision:** Masaharu Tsubokura.

**Validation:** Yuya Kashiwazaki.

**Writing – original draft:** Hitomi Matsunaga.

**Writing – review & editing:** Makiko Orita, Thierry Schneider, Masaharu Tsubokura, Noboru Takamura.

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
