## [Decision Letter · Decision Letter 0]

3 Nov 2025

PONE-D-25-11888

Recognition of the younger generation a year after the discharge of treated water from the Fukushima Daiichi Nuclear Power Station

PLOS ONE

Dear Dr. Matsunaga,

Thank you for submitting your manuscript to PLOS ONE. After careful consideration, we feel that it has merit but does not fully meet PLOS ONE’s publication criteria as it currently stands. Therefore, we invite you to submit a revised version of the manuscript that addresses the points raised during the review process.

We look forward to receiving your revised manuscript.

Kind regards,

Shervin Jamshidi

Academic Editor

PLOS ONE

Journal Requirements:

2. Please include a complete copy of PLOS’ questionnaire on inclusivity in global research in your revised manuscript. Our policy for research in this area aims to improve transparency in the reporting of research performed outside of researchers’ own country or community. The policy applies to researchers who have travelled to a different country to conduct research, research with Indigenous populations or their lands, and research on cultural artefacts. The questionnaire can also be requested at the journal’s discretion for any other submissions, even if these conditions are not met.

Please find more information on the policy and a link to download a blank copy of the questionnaire here: https://journals.plos.org/plosone/s/best-practices-in-research-reporting.

Please upload a completed version of your questionnaire as Supporting Information when you resubmit your manuscript.

5. Thank you for stating the following in your Competing Interests section:  “No”

6. We note that you have indicated that there are restrictions to data sharing for this study. For studies involving human research participant data or other sensitive data, we encourage authors to share de-identified or anonymized data. However, when data cannot be publicly shared for ethical reasons, we allow authors to make their data sets available upon request. For information on unacceptable data access restrictions, please see http://journals.plos.org/plosone/s/data-availability#loc-unacceptable-data-access-restrictions.

7. Your ethics statement should only appear in the Methods section of your manuscript. If your ethics statement is written in any section besides the Methods, please move it to the Methods section and delete it from any other section. Please ensure that your ethics statement is included in your manuscript, as the ethics statement entered into the online submission form will not be published alongside your manuscript.

9. We note that there is identifying data in the Supporting Information file < Appendix 1 Age of the study participants.docx>. Due to the inclusion of these potentially identifying data, we have removed this file from your file inventory. Prior to sharing human research participant data, authors should consult with an ethics committee to ensure data are shared in accordance with participant consent and all applicable local laws.

-Location data

Please remove or anonymize all personal information, ensure that the data shared are in accordance with participant consent, and re-upload a fully anonymized data set. Please note that spreadsheet columns with personal information must be removed and not hidden as all hidden columns will appear in the published file.

**Additional Editor Comments:**

Authors should carefully address the whole reviewers' comments before submitting their revised manuscript.

Reviewers' comments:

Reviewer's Responses to Questions

**Comments to the Author**

1. Is the manuscript technically sound, and do the data support the conclusions?

Reviewer #1: Yes

Reviewer #2: No

2. Has the statistical analysis been performed appropriately and rigorously?

Reviewer #1: No

Reviewer #2: No

3. Have the authors made all data underlying the findings in their manuscript fully available?

Reviewer #1: No

Reviewer #2: No

4. Is the manuscript presented in an intelligible fashion and written in standard English?

Reviewer #1: Yes

Reviewer #2: Yes

Reviewer #1: Overall Assessment:

This manuscript presents a timely and socially important study on the awareness and acceptance of the discharge of treated water (DTW) from the Fukushima Daiichi Nuclear Power Station (FDNPS) among Japanese university students, one year after the discharge began. Based on responses from 1,453 students, the analysis is generally sound and offers valuable insight into how information sources, trust, and demographic factors influence DTW acceptance. However, several revisions are necessary to improve the manuscript’s clarity, methodological transparency, and interpretative strength.

Major Comments:

1. Clarification of Novelty and Significance (Introduction):

The manuscript would benefit from a clearer articulation of its novelty, particularly the rationale for focusing on the younger generation. Since this group will bear long-term responsibility for national and environmental policy, prior literature on youth engagement in environmental issues or risk communication should be cited to frame the importance of their perceptions.

2. Sampling Bias and Generalizability (Methods):

The authors state that the sample was drawn from four universities, but more detail is needed about potential sampling bias—especially with respect to students’ academic backgrounds. For example, students in medicine, nursing, or engineering may have greater baseline knowledge of radiation, which could influence their acceptance of DTW.

In addition, please provide more detailed information about the geographic location of the universities outside Fukushima Prefecture. Since the treated water is being discharged into the Pacific Ocean, perceptions may differ between coastal areas facing the Pacific and more inland regions. Without this information, it is difficult to fully assess the effect of regional proximity to the discharge site on participants’ responses.

3. Logistic Regression Model Interpretation:

In Table 3, sex was statistically significant in the chi-square analysis but not in the logistic regression model. This inconsistency should be discussed in the manuscript—perhaps in terms of confounding variables or multicollinearity.

Moreover, the current model omits potentially important predictors such as residential area (Fukushima vs. non-Fukushima) and age, both of which may shape attitudes toward DTW. Including these variables in the regression model may enhance its explanatory power.

If the data are available, it would also be helpful to incorporate students’ academic discipline (e.g., science vs. humanities) into the model. Educational background may influence scientific literacy and thus affect DTW perception and acceptance.

4. Transparency of Survey Instrument:

To improve the study’s transparency and reproducibility, the full questionnaire should be included—either in an appendix or as supplementary material. While the Methods section describes the survey’s structure in broad terms, access to the full set of questions would allow readers to better assess construct validity, especially for subjective items like “trust in the Japanese government” or “public calm.” Including both the original Japanese and an English translation would be ideal.

5. Interpretation of Media Influence and Lecture Exposure:

The finding that lecture attendance did not significantly affect DTW acceptance requires deeper interpretation. It would be helpful for the authors to discuss whether this was due to the content, quality, or timing of the lectures.

Crucially, it is unclear whether all participants were equally exposed to lectures on DTW. If lectures were only given to some students, the observed lack of effect may reflect variation in exposure rather than ineffectiveness. This should be clarified in the Methods section, and the implications discussed in the limitations.

Additionally, the fact that passive media sources (TV/newspapers) were more strongly associated with DTW acceptance suggests that mass communication may be more effective than formal educational interventions. This has important implications for public outreach strategies.

Minor Comments:

Terminology Consistency:

The terms "contaminated water" and "treated water" appear to be used interchangeably at times. Please clearly define both terms early in the manuscript and ensure consistent usage, given the public sensitivity surrounding their distinction.

Reviewer #2: Keywords require revision.

The respondents of questionnaire are all college students. It can not be a good representative for younger generation, because a generation includes both illiterate and literate people or consists of a wide range of literacy level.

It seems the comparison between younger and older people is based on 21 years old. This is not what we understand from the title, abstract, or conclusion. Line 259 can not be verified by the data in manuscript.

Obviously, younger people in general has relatively less concerns about their health in comparison with older people. I wonder what are are the perceptions of people with 30 years old or higher?

Manuscript requires proper illustrations to reflect their results. Tables 2 and 3 have the potential for be in box-plot figures.

Authors must demonstrate the results of all required statistical analysis: ANOVA or Mann-whithney, Mendel test, alpha cronbach, Pearson regression, etc. The current manuscript is more like a Poll than a scientific paper with professional scientific analysis.

I wonder what other independent factors, like being employed, having families, and wealth are effective on the acceptance on wastewater discharge? Can they represent how much conservative or ignorant are people to this event?

In the current format, I can not approve the validity of results, unless authors improve and verify their findings with solid methodology and statistical analysis. This manuscript raises more questions than answering to the ambiguities. Therefore, it requires major revision.

**Do you want your identity to be public for this peer review?** For information about this choice, including consent withdrawal, please see our Privacy Policy

Reviewer #1: No

Reviewer #2: No

---

## [Author Response · Author response to Decision Letter 1]

19 Nov 2025

Thank you for your valuable comments and suggestions.

Based on reviewers' suggestions, we have revised our manuscript accordingly. Our point-by-point responses are as follows:

Reviewer #1: Overall Assessment:

This manuscript presents a timely and socially important study on the awareness and acceptance of the discharge of treated water (DTW) from the Fukushima Daiichi Nuclear Power Station (FDNPS) among Japanese university students, one year after the discharge began. Based on responses from 1,453 students, the analysis is generally sound and offers valuable insight into how information sources, trust, and demographic factors influence DTW acceptance. However, several revisions are necessary to improve the manuscript’s clarity, methodological transparency, and interpretative strength.

We appreciate your positive feedback and valuable suggestions.

Major Comments:

1. Clarification of Novelty and Significance (Introduction):

The manuscript would benefit from a clearer articulation of its novelty, particularly the rationale for focusing on the younger generation. Since this group will bear long-term responsibility for national and environmental policy, prior literature on youth engagement in environmental issues or risk communication should be cited to frame the importance of their perceptions.

Thank you for your advice. In line with your suggestion, I have added previous studies on youth engagement in environmental issues or risk communication to the introduction session (L.56-62).

2. Sampling Bias and Generalizability (Methods)

The authors state that the sample was drawn from four universities, but more detail is needed about potential sampling bias—especially with respect to students’ academic backgrounds. For example, students in medicine, nursing, or engineering may have greater baseline knowledge of radiation, which could influence their acceptance of DTW.

I appreciate your advice. Following your suggestions, I have added students’ academic backgrounds to Appendix 1 and the analysis.

In addition, please provide more detailed information about the geographic location of the universities outside Fukushima Prefecture. Since the treated water is being discharged into the Pacific Ocean, perceptions may differ between coastal areas facing the Pacific and more inland regions. Without this information, it is difficult to fully assess the effect of regional proximity to the discharge site on participants’ responses.

Furthermore, based on your suggestion, I added general information regarding the geographic distribution of universities outside Fukushima Prefecture to the Study participants of the Methods section (L.71-73).

3. Logistic Regression Model Interpretation:

In Table 3, sex was statistically significant in the chi-square analysis but not in the logistic regression model. This inconsistency should be discussed in the manuscript—perhaps in terms of confounding variables or multicollinearity.

Thank you for your advice. Based on your suggestion, we added an illustration that sex was not significant in the logistic regression model in the results( L169-170 and 215-217).

Moreover, the current model omits potentially important predictors such as residential area (Fukushima vs. non-Fukushima) and age, both of which may shape attitudes toward DTW. Including these variables in the regression model may enhance its explanatory power.

Thank you for your advice. Although area (Fukushima vs. non-Fukushima) and age appeared to be factors influencing acceptance of DTW, the chi-square test comparing acceptance vs non-acceptance did not show any significant differences, so we did not include them in the logistic model. As a trial, I constructed a model including these two factors, but no significant associations were observed. Please refer to the table below (next page) for details.

If the data are available, it would also be helpful to incorporate students’ academic discipline (e.g., science vs. humanities) into the model. Educational background may influence scientific literacy and thus affect DTW perception and acceptance.

We greatly appreciate your valuable advice. Based on your suggestion, upon incorporating Major(academic background) into the model, a significant association was identified. Consequently, the findings have been adapted to the paper.

Table 3. Independent factors associated with acceptance of DTW from FDNPS

OR (95% CI) p-value

Sex Female/Male 0.897 (0.648–1.242) 0.512

Age < 21/� 21 0.989 (0.933–1.047) 0.697

Living area Inside/Outside 0.988 (0.684–1.427) 0.949

Major Science/Humanities 1.505 (1.004–2.236) 0.048

The experience of collecting information or knowledge about the DTW Yes/No 1.554 (1.119–2.169) 0.009

Ability to explain the difference between contaminated water and treated water Yes/No 1.832 (1.319–2.808) <0.001

The accuracy of information about DTW provided by the Japanese government Yes/No 1.683 (1.365–2.075) <0.001

The calm decision about DTW in public opinion Yes/No 1.833 (1.257–2.674) 0.002

Concerns about consuming seafood from Fukushima No/Yes 3.742 (2.7708–5.225) <0.001

Note. Logistic regression analysis, reference; acceptance of DTW from FDNPS; OR, odds ratio; CI, confidence interval; FDNPS, Fukushima Daiichi Nuclear Power Station; DTW, discharge of treated water

4. Transparency of Survey Instrument:

To improve the study’s transparency and reproducibility, the full questionnaire should be included—either in an appendix or as supplementary material. While the Methods section describes the survey’s structure in broad terms, access to the full set of questions would allow readers to better assess construct validity, especially for subjective items like “trust in the Japanese government” or “public calm.” Including both the original Japanese and an English translation would be ideal.

Thank you for your advice on improving the clarity of the paper. Based on your suggestions, I have added the questionnaire in both English and Japanese to Appendix 2.

5. Interpretation of Media Influence and Lecture Exposure:

The finding that lecture attendance did not significantly affect DTW acceptance requires deeper interpretation. It would be helpful for the authors to discuss whether this was due to the content, quality, or timing of the lectures.

Crucially, it is unclear whether all participants were equally exposed to lectures on DTW. If lectures were only given to some students, the observed lack of effect may reflect variation in exposure rather than ineffectiveness. This should be clarified in the Methods section, and the implications discussed in the limitations.

I sincerely appreciate your valuable advice. In this study, we were only able to confirm whether participants had attended or experienced a lecture on DTW. This has been added as a limitation section (L278-283).

Additionally, the fact that passive media sources (TV/newspapers) were more strongly associated with DTW acceptance suggests that mass communication may be more effective than formal educational interventions. This has important implications for public outreach strategies.

We sincerely appreciate your valuable insights. The points you provided have been added to the discussion session (L229-232).

Minor Comments:

Terminology Consistency:

The terms "contaminated water" and "treated water" appear to be used interchangeably at times. Please clearly define both terms early in the manuscript and ensure consistent usage, given the public sensitivity surrounding their distinction.

Thank you for your creative advice. Based on your suggestions, we have added a clear definition of both terms in the Introduction session (L37-41).

Reviewer #2:

Keywords require revision.

The respondents of questionnaire are all college students. It can not be a good representative for younger generation, because a generation includes both illiterate and literate people or consists of a wide range of literacy level. It seems the comparison between younger and older people is based on 21 years old. This is not what we understand from the title, abstract, or conclusion.

Thank you for your valuable comment. Based on your suggestion, we revised the keywords and the expression “the younger generation” to “University student”. Furthermore, we add “in Japan” because Japan’s literacy rate is close to 100%.

Line 259 can not be verified by the data in manuscript. Obviously, younger people in general has relatively less concerns about their health in comparison with older people. I wonder what are are the perceptions of people with 30 years old or higher?

Thank you for your insightful comment. As you pointed out, younger people in general have relatively fewer concerns about their health compared to older people. However, since this study focused on university students, the population aged 30 and above accounted for only about 2% of the participants (kindly refer to Appendix 2), which posed limitations for the analysis.

Manuscript requires proper illustrations to reflect their results. Tables 2 and 3 have the potential for be in box-plot figures.

Thank you for your valuable comments. Table 2 presents the results of the chi-square test, and Table 3 shows the logistic regression analysis result. Although it is possible to create a box-plot figure for Table 2, the purpose of Table 2 is to demonstrate that there is no difference in the proportions of 'Accept' and 'Do not Accept' depending on the source of DTW information, rather than to indicate differences in median or mean values. Furthermore, Table 3 cannot be converted into a figure. We apologize for not being able to fully follow your creative advice.

Authors must demonstrate the results of all required statistical analysis: ANOVA or Mann-whithney, Mendel test, alpha cronbach, Pearson regression, etc. The current manuscript is more like a Poll than a scientific paper with professional scientific analysis.

Thank you for your creative advice. This study compares results based on proportions rather than medians or means. For analyzing associations, we employed logistic regression instead of correlation analysis. This paper presents a simple approach based on the questionnaire data obtained; therefore, we did not adopt approaches that present the same results through multiple figures or analyses. This is the limitation of our questionnaire. We regret that we were unable to fully implement your suggestion.

I wonder what other independent factors, like being employed, having families, and wealth are effective on the acceptance on wastewater discharge? Can they represent how much conservative or ignorant are people to this event? In the current format, I can not approve the validity of results, unless authors improve and verify their findings with solid methodology and statistical analysis. This manuscript raises more questions than answering to the ambiguities. Therefore, it requires major revision.

Thank you for your advice. Since this questionnaire targeted university students, basic attributes such as being employed, having families, and wealth were not collected. Regarding the items you pointed out, we plan to revise the participants and the questionnaire and clarify them in the next survey with the new questionnaire. Thank you also for your suggestions concerning the future direction of the study. Based on your suggestion, we added information regarding basic attributes in the limitation section.

---

## [Decision Letter · Decision Letter 1]

4 Feb 2026

Dear Dr. Matsunaga,

Thank you for submitting your manuscript to PLOS ONE. After careful consideration, we feel that it has merit but does not fully meet PLOS ONE’s publication criteria as it currently stands. Therefore, we invite you to submit a revised version of the manuscript that addresses the points raised during the review process.

We look forward to receiving your revised manuscript.

Kind regards,

Sakae Kinase, Ph.D.

Academic Editor

PLOS One

Journal Requirements:

Additional Editor Comments (if provided):

Your revised manuscript has been reviewed by two additional referees. Please carefully consider the comments and suggestions from the referees and resubmit, as soon as possible, an amended version of the manuscript, including confirming conflict of interest declaration, prepared in accordance with the Instructions to Authors. It is anticipated that the manuscript will be accepted in its amended form for publication in PLOS One.

Reviewers' comments:

Reviewer's Responses to Questions

**Comments to the Author**

Reviewer #1: All comments have been addressed

Reviewer #2: (No Response)

Reviewer #3: All comments have been addressed

Reviewer #4: (No Response)

2. Is the manuscript technically sound, and do the data support the conclusions?

Reviewer #1: Yes

Reviewer #2: No

Reviewer #3: Yes

Reviewer #4: Partly

3. Has the statistical analysis been performed appropriately and rigorously?

Reviewer #1: Yes

Reviewer #2: No

Reviewer #3: Yes

Reviewer #4: Yes

4. Have the authors made all data underlying the findings in their manuscript fully available?

Reviewer #1: Yes

Reviewer #2: No

Reviewer #3: Yes

Reviewer #4: Yes

5. Is the manuscript presented in an intelligible fashion and written in standard English?

Reviewer #1: Yes

Reviewer #2: Yes

Reviewer #3: Yes

Reviewer #4: Yes

Reviewer #1: (No Response)

Reviewer #2: Regarding the revised manuscript, response letter, and clarified method and results by responses, I can now admit that this manuscript has lack of scientific results for publication:

1- This manuscript is limited to a very specific society (university students) in Japan. This group under survey has very limited variety in social characteristics (age, job, literacy, families, etc.). The independent variables are limited to >21 or <21 age, their majors, and methods for information collecting. This group can not represent any other similar groups in Japan or other countries. Obviously, the results are not suitable with scientific perspective, as these outcomes can not be extended to any other case, group, or society.

2- As explained by authors, the method for analyzing the outcomes is limited to some comparisons by percentage (%) like line 128-153. In other words, the statistical analysis was more focused on frequency (like a poll), while authors could briefly explain these outcomes by illustrations.

3- In a nutshell, the method, including surveyed group and statistical analysis, have nothing special or innovative that make the manuscript different from a poll or a typical survey. The discussion also lacks scientific descriptions about the outcomes.

4- Authors could use new approaches in survey, conceptual modeling, or etc. to strengthen their manuscript. But, unfortunately, the current revised manuscript can not meet the demands of a world-wide replicable scientific research.

Reviewer #3: The manuscript reports on university students’ awareness of the release of ALPS-treated water from the Fukushima Daiichi Nuclear Power Plant. As this is a revised version, the following comments focus on the authors’ responses to the reviewers’ comments.

The authors have provided complete and appropriate responses to the comments raised by Reviewer #1. With respect to Reviewer #2, the authors have adequately addressed the major criticisms by acknowledging the study’s limitations and revising the title accordingly.

Although Reviewer #2 characterized the study as resembling a poll, the analysis is clearly focused on university students, a population with sufficient prior knowledge and intellectual maturity to engage with the topic. From this perspective, the study does not appear to be a simple poll. While expanding the target population to include the general public could be valuable, this point can reasonably be considered a topic for future research, as noted by the authors. The awareness of university students, who are expected to play leading roles in the next generation, represents a meaningful and justified focus.

Reviewer #4: General comment

This study focuses on acceptance of DTW, mainly in terms of its scientific safety. However, there is another critical argument on fair and transparent decision-making process of DTW with stakeholders, including fishery sector. Though the questionnaire includes a question for “calm decision by public”, but no question is asked for acceptance of policy decision making process nor access to information on procedures of governmental decision making, conflicts with local stakeholders, etc. The paper might justify DTW, by supportive opinion by young generation, who may not be informed about conflicts with stakeholders, as such aspect is (intentionally or un-intentionally) excluded from the questionnaire. In the view of the reviewer, more essential point of DTW is lack of stakeholder involvement in decision making process, rather than scientific safety. The questionnaire could have been designed to include acceptance not only of scientific safety, but also social fairness, international justice, etc.

Specific comments

1. Response to the comment on literacy by the Reviewer 2 seems to be inappropriate. Authors responded in terms of language/character literacy, the reviewer’s comment addressed scientific literacy, in my understanding.

2. The comment#1 above is interlinked with the comment #2 by the Reviewer 1 on sampling bias. Difference in acceptance between majors (natural science/humanities) is mentioned in conclusion, but it should be noted that the percentage of respondents between these two majors (83.4% vs. 16.6%) is significantly different from total population of graduate students. Student population of humanity is much larger than student population in natural sciences, in Japan. Such bias could have been avoided by more careful sampling design. This comment is also related to general comment above. The study seems to focus too much on natural scientific aspect, and underestimates the importance of humanity.

**Do you want your identity to be public for this peer review?** For information about this choice, including consent withdrawal, please see our Privacy Policy

Reviewer #1: **Yes:** Takeshi Takahaashi

Reviewer #2: No

Reviewer #3: No

Reviewer #4: No

---

## [Author Response · Author response to Decision Letter 2]

12 Feb 2026

Thank you for your valuable comments and suggestions.

Based on reviewers' suggestions, we have revised our manuscript accordingly. Our point-by-point responses are as follows:

Reviewer #2: Regarding the revised manuscript, response letter, and clarified method and results by responses, I can now admit that this manuscript has lack of scientific results for publication:

1- This manuscript is limited to a very specific society (university students) in Japan. This group under survey has very limited variety in social characteristics (age, job, literacy, families, etc.). The independent variables are limited to >21 or <21 age, their majors, and methods for information collecting. This group can not represent any other similar groups in Japan or other countries. Obviously, the results are not suitable with scientific perspective, as these outcomes can not be extended to any other case, group, or society.

We appreciate your thoughtful comments regarding the generalizability of our study. As you pointed out, the present research targeted university students in Japan, who represent a relatively homogeneous population in terms of age and social characteristics. We fully acknowledge that this sampling frame limits the extent to which our findings can be generalized to other populations.

In this study, we specifically aimed to examine the perceptions of Japanese university students, who represent the next generation in a country currently DTW into the ocean. As described in the Limitations section, there is a potential sampling bias because the participants were drawn from two universities inside Fukushima Prefecture and two universities outside the prefecture (p. 19, L266-267).

At present, we do not have comparable data from university students or other populations in other countries; therefore, we cannot make direct cross-national comparisons. Acceptance of DTW may differ across cultural or societal contexts. In response to your comment, we have added this point explicitly to the Limitation sections of the revised manuscript (p. 19, L269-274). We believe that future research with more diverse samples, including international populations, will be essential to broaden the scientific contribution of this line of research.

We thank you again for this valuable suggestion, which has helped us to clarify the scope and limitations of our study.

2- As explained by authors, the method for analyzing the outcomes is limited to some comparisons by percentage (%) like line 128-153. In other words, the statistical analysis was more focused on frequency (like a poll), while authors could briefly explain these outcomes by illustrations.

We appreciate the valuable comment. The section described in lines 128–153 refers to the “Comparisons of each factor accepting DTW from the FDNPS,” and all corresponding results are summarized in Table 1. Table 1 presents both (1) the descriptive statistics of each factor and (2) the results of the chi-square tests comparing the two group participants who accepted DTW from the FDNPS and those who did not. The analyses provided in this study rely on frequency‑based statistical comparisons.

3- In a nutshell, the method, including surveyed group and statistical analysis, have nothing special or innovative that make the manuscript different from a poll or a typical survey. The discussion also lacks scientific descriptions about the outcomes.

We sincerely appreciate the reviewer’s thoughtful insights regarding the methodological rigor and scientific contributions of the manuscript. Our study aims to provide empirical evidence regarding the acceptance of DTW from the FDNPS within a specific population, and thus employs a population-based survey design that is commonly used in risk perception and public acceptance research. While the method itself may not be innovative in terms of technique, it was carefully designed to obtain reliable, context‑specific evidence that has not previously been investigated in this region.

4- Authors could use new approaches in survey, conceptual modeling, or etc. to strengthen their manuscript. But, unfortunately, the current revised manuscript can not meet the demands of a world-wide replicable scientific research.

We thank the reviewer for this constructive suggestion. We acknowledge that advanced methodologies—such as conceptual modeling, structural equation modeling, or novel survey frameworks—could further enhance generalizability and allow for broader replication. At the same time, the primary objective of this study was to obtain baseline empirical data from a specific local context, where no comparable dataset existed. For this purpose, a conventional survey design was considered the most appropriate and feasible approach.

Nonetheless, following the reviewer’s recommendation, we have added a paragraph in the Limitations section explicitly addressing the scope of the methodology and outlining potential future directions, including the incorporation of conceptual modeling, longitudinal follow-ups, and expanded sampling strategies to enhance international replicability (p. 21, L305-312). We appreciate the reviewer’s perspective and believe these additions improve the transparency and future applicability of the research.

Reviewer #3: The manuscript reports on university students’ awareness of the release of ALPS-treated water from the Fukushima Daiichi Nuclear Power Plant. As this is a revised version, the following comments focus on the authors’ responses to the reviewers’ comments.

The authors have provided complete and appropriate responses to the comments raised by Reviewer #1. With respect to Reviewer #2, the authors have adequately addressed the major criticisms by acknowledging the study’s limitations and revising the title accordingly.

Although Reviewer #2 characterized the study as resembling a poll, the analysis is clearly focused on university students, a population with sufficient prior knowledge and intellectual maturity to engage with the topic. From this perspective, the study does not appear to be a simple poll. While expanding the target population to include the general public could be valuable, this point can reasonably be considered a topic for future research, as noted by the authors. The awareness of university students, who are expected to play leading roles in the next generation, represents a meaningful and justified focus.

We sincerely appreciate your careful review of our revised manuscript and your constructive comments.

Reviewer #4: General comment

This study focuses on acceptance of DTW, mainly in terms of its scientific safety. However, there is another critical argument on fair and transparent decision-making process of DTW with stakeholders, including fishery sector. Though the questionnaire includes a question for “calm decision by public”, but no question is asked for acceptance of policy decision making process nor access to information on procedures of governmental decision making, conflicts with local stakeholders, etc. The paper might justify DTW, by supportive opinion by young generation, who may not be informed about conflicts with stakeholders, as such aspect is (intentionally or un-intentionally) excluded from the questionnaire. In the view of the reviewer, more essential point of DTW is lack of stakeholder involvement in decision making process, rather than scientific safety. The questionnaire could have been designed to include acceptance not only of scientific safety, but also social fairness, international justice, etc.

We sincerely appreciate the reviewer’s insightful and important comments regarding the decision‑making processes surrounding DTW and the role of stakeholder involvement. We fully agree that fair, transparent, and participatory decision‑making—particularly with stakeholders such as the fishery sector—is a critical dimension of the DTW issue, and that scientific safety represents only one aspect of broader public acceptance.

We acknowledge that the present questionnaire primarily focused on perceptions related to the scientific safety of DTW. Although one item addressed “the calm decision about DTW in public,” the survey did not include explicit questions about acceptance of the governmental decision‑making process, access to information, conflict with local stakeholders, or views on procedural fairness and justice. We agree that these aspects represent essential components of social acceptance and are highly relevant to the broader policy context.

To address this valuable point, we have added a detailed explanation in the Limitations section clarifying that:

・The survey did not assess procedural fairness, stakeholder involvement, or perceived legitimacy of the decision‑making process; therefore, it reflects only one dimension of acceptance (scientific safety–related perceptions).

・Future studies should incorporate questions on social fairness, international justice, and stakeholder engagement to more comprehensively evaluate acceptance of DTW. (p. 20-21, L297-302)

We highlight this explicitly to avoid any unintended misinterpretation of the study’s scope or conclusions.

We appreciate the reviewer’s thoughtful guidance, which has helped us clarify the intended scope of the study and ensure transparent representation of its limitations.

Specific comments

1. Response to the comment on literacy by the Reviewer 2 seems to be inappropriate. Authors responded in terms of language/character literacy, the reviewer’s comment addressed scientific literacy, in my understanding.

We thank the reviewer for pointing out this misunderstanding. We acknowledge that our previous response did not fully address the reviewer’s intended meaning. The original comment was referring not to language or character literacy, but to scientific literacy, which plays an important role in how individuals interpret information related to DTW.

In the revised manuscript, we have corrected this misunderstanding and clarified that differences in scientific literacy, rather than general reading/writing literacy, may influence respondents’ understanding of technical information on radiation, environmental monitoring, and safety standards. We added this clarification in the Limitations section, and we explicitly acknowledge that our survey did not include a direct measure of scientific literacy. (p. 21, L302-305)

2. The comment#1 above is interlinked with the comment #2 by the Reviewer 1 on sampling bias. Difference in acceptance between majors (natural science/humanities) is mentioned in conclusion, but it should be noted that the percentage of respondents between these two majors (83.4% vs. 16.6%) is significantly different from total population of graduate students. Student population of humanity is much larger than student population in natural sciences, in Japan. Such bias could have been avoided by more careful sampling design. This comment is also related to general comment above. The study seems to focus too much on natural scientific aspect, and underestimates the importance of humanity.

We appreciate this important comment regarding sampling bias and the representation of academic majors. We agree that the proportion of respondents from natural sciences and humanities in our sample does not reflect the national distribution of graduate students in Japan. As the reviewer notes, students in natural sciences constitute a larger share of the overall student population. This imbalance was influenced by differences in response rates among departments, and we recognize that this may affect the generalizability of our findings.

To address this concern, we have revised the Limitations section to explicitly acknowledge the disproportionate representation of natural science majors and their potential influence on observed acceptance levels. (p. 21, L281-286).

We also clarify that the difference observed between natural science and humanities majors should be interpreted with caution, given the uneven sample distribution.

Moreover, we fully agree with the reviewer’s point that social and humanities perspectives are essential for understanding the broader societal dimensions of DTW acceptance, including ethics, social justice, procedural fairness, and stakeholder engagement. We have strengthened the Limitation to emphasize that acceptance of DTW cannot be understood solely through a natural‑science lens, and that future studies should intentionally design sampling strategies to include a more balanced representation of humanities and social‑science students, or other populations with diverse epistemological backgrounds.

We appreciate the reviewer’s insight, which has helped us clarify the scope of our study and improve transparency regarding sampling limitations.

---

## [Editor Report · Decision Letter 2]

23 Feb 2026

Recognition of Japanese university students one year after the discharge of treated water from the Fukushima Daiichi Nuclear Power Station

PONE-D-25-11888R2

Dear Dr. Matsunaga,

We’re pleased to inform you that your manuscript has been judged scientifically suitable for publication and will be formally accepted for publication once it meets all outstanding technical requirements.

Kind regards,

Sakae Kinase, Ph.D.

Academic Editor

PLOS One

Additional Editor Comments (optional):

I am pleased to inform you that your revised manuscript has been accepted for publication in PLOS One as a Research Article.
---

## [Editor Report · Acceptance letter]

PONE-D-25-11888R2

PLOS One

Dear Dr. Matsunaga,

I'm pleased to inform you that your manuscript has been deemed suitable for publication in PLOS One. Congratulations! Your manuscript is now being handed over to our production team.

Kind regards,

on behalf of

Professor Sakae Kinase

Academic Editor

PLOS One